# Is There a Dose–Response Relationship between High-Intensity Interval Exercise (HIIE) Intensity and Affective Valence? Analysis of Three HIIE Sessions Performed with Different Amplitudes

**DOI:** 10.3390/ijerph20032698

**Published:** 2023-02-02

**Authors:** Michel Oliveira Silva, Tony Meireles Santos, Allan Inoue, Lucas Eduardo Rodrigues Santos, Weydyson de Lima do Nascimento Anastácio, Eduardo Lattari, Bruno Ribeiro Ramalho Oliveira

**Affiliations:** 1Exercise and Sport Sciences Postgraduate Program, Rio de Janeiro State University, Rio de Janeiro 20550-013, RJ, Brazil; 2Physical Education Department, Pernambuco Federal University, Recife 50670-901, PE, Brazil; 3Physical Activity Sciences Postgraduate Program, Salgado de Oliveira University (UNIVERSO), Niterói 24030-060, RJ, Brazil; 4Physical Education and Sports Department, Federal Rural University of Rio de Janeiro, Seropédica 23890-000, RJ, Brazil

**Keywords:** affective valence, HIIE, VICE, perceived exertion

## Abstract

The inverse relationship between exercise intensity and affective valence is well established for continuous exercise but not for high-intensity interval exercise (HIIE). The objective was to verify the dose–response relationship between exercise intensity and affective valence in HIIE sessions. Eleven young men underwent a vigorous-intensity continuous exercise (VICE) and three HIIE sessions at the same average intensity (70% of peak power—W_Peak_) and duration (20 min) but with different amplitudes: 10 × [1 min at 90% W_Peak_/1 min at 50% W_Peak_]—HIIE-90/50; 10 × [1 min at 100% W_Peak_/1 min at 40% W_Peak_]—HIIE-100/40; 10 × [1 min at 110% W_Peak_/1 min at 30% W_Peak_]—HIIE-110/30. During the exercise sessions, psychophysiological variables were recorded (VO_2_, VCO_2_, heart rate, perceived exertion CR10, and Feeling Scale (FS)). Higher correlations were found between CR10 and FS for all conditions (VICE = −0.987; HIIE-90/50 = −0.873; HIIE-100/40 = −0.908; HIIE-110/30 = −0.948). Regarding the physiological variables, the %HR_Max_ presented moderate inverse correlations with FS for all exercise conditions (VICE = −0.867; HIIE-90/50 = −0.818; HIIE-100/40 = −0.837; HIIE-110/30 = −0.828) while the respiratory variables (%VO_2Peak_ and %VCO_2Peak_) presented low-to-moderate correlations only for VICE, HIIE-90/50, and HIIE-100/40 (ranging from −0.523 to −0.805). Poor correlations were observed between the %VO_2Peak_ (r = −0.293) and %VCO_2Peak_ (r = −0.020) with FS. The results indicated that perceived exertion is more sensible than physiological variables to explain the intensity–affective valence relationship in HIIE sessions. RPE should be used for HIIE prescription with a focus on affect.

## 1. Introduction

There is growing interest from the scientific community regarding the applicability of affective responses to exercise [1]. Specifically, an inverse relationship between affective valence and continuous exercise intensity has been demonstrated [2]. This relationship was not clearly shown for high-intensity interval exercise (HIIE), possibly due to the higher number of configuration variables involved in the HIIE prescription [3,4]. Therefore, attaining optimal affective responses in HIIE is challenging for scientists and coaches to establish the dose–response relationship between different HIIE protocols and affective valence.

To our knowledge, three studies investigated the relationship between the HIIE intensity and the affective valence [5,6,7]. Ramalho Oliveira, Viana, Pires, Junior Oliveira, and Santos [5] showed an inverse correlation between the ratings of perceived exertion (RPE) and the affective valence in a single HIIE session (r = −0.82). Similarly, Frazao, de Farias Junior, Dantas, Krinski, Elsangedy, Prestes, Hardcastle, and Costa [6] also found an inverse correlation between affective valence and RPE (r = −0.56). More recently, Farias-Junior, Browne, Astorino, and Costa [7] presented large inverse correlations between affective valence and RPE, especially at the end of the HIIE session (RPE–stimulus period: r = −0.82; RPE–recovery period: r = −0.68). These studies performed HIIE sessions at an average intensity of 85% of respiratory compensation point [5], and 60% of maximal speed was obtained during the maximal test [6,7] with stimulus intensity above the metabolic thresholds. In this context, Ramalho Oliveira, Viana, Pires, Junior Oliveira, and Santos [5] suggested that the affective responses seem to be modulated not only by the exercise intensity but also by how the individuals perceived this intensity.

Despite the scientific progress presented by these studies [5,6,7], thedose–response relationship between HIIE intensity and affective valence was not demonstrated, considering that only one HIIE session was performed and, therefore, thedose–response relationship could not be tested for different HIIE sessions. In this sense, two HIIE sessions conducted at the same stimulus and recovery intensities may be configured with different stimulus and recovery duration, resulting in different physiological responses [8] and, possibly, affective responses considering the interoceptive cues which could modulate affect, especially in high-intensity efforts [9]. It was previously proposed that the amplitude, described by Billat [10], may be used to interpret the impact of different HIIE sessions by analyzing only one variable [3]. However, even applying HIIE sessions with different amplitudes [3], the authors did not investigate thedose–response of HIIE intensities and affective responses. Therefore, it is necessary to investigate the relationship between psychophysiological markers of exercise intensity and affective valence to establish theirdose–response effect.

Based on this premise, the objective of the present study was to verify the dose–response relationship between exercise intensity and affective valence in three HIIE sessions performed with different amplitudes but at the same average intensity and total time. On the basis of previous findings [5,6,7], we hypothesized that a dose–response relationship between HIIE intensity and affective valence might be established using the RPE as the intensity variable.

## 2. Materials and Methods

### 2.1. Participants

The present study is part of a research project, and its methodological procedures were previously described [3]. Twelve men participated in the present study; however, one individual failed to complete the experimental sessions. Therefore, 11 men were included in the final analyses. Participants aged 18–35 years old and at low risk for cardiovascular disease [11], physically active or not, were included in the study. Individuals with musculoskeletal disorders or with a diagnosis of mental disorders were excluded from the study. Individuals with resting blood pressure ≥139/89 mmHg in three consecutive measurements were also excluded from the study. The written consent form approved by the institutional ethics committee (# 1.385.003) was presented and signed by the participants.

### 2.2. Experimental Design

In order to establish the relationship between exercise intensity and affective valence, four internal load indicators were chosen to be included in the analysis. The oxygen consumption (VO_2_), dioxide carbon output (VCO_2_), and heart rate (HR) were used as physiological indicators of the exercise intensity, while the Category Ratio Scale (CR10) was used as a psychological indicator of the exercise intensity. The VO_2_ and VCO_2_ were included considering their relationship with aerobic and anaerobic demands during exercise, respectively [12]. The HR is an indicator of cardiovascular load during exercise, and CR10 is a psychological indicator of the internal load. Therefore, the exercise intensity was quantified by different internal load variables to determine the dose–response relationship between HIIE intensity and affective valence. The affective valence was measured using the Feeling Scale (FS) [13]. On these methodological premises, participants were asked to perform five visits on a cycle ergometer (RacerMate CompuTrainer, Seattle, WA, USA). During the first visit, participants signed the consent form and completed the risk stratification questionnaire [11]. Then, the resting heart rate (HR), blood pressure, and anthropometric measurements were taken. After these procedures, participants completed the maximal test in which HR and respiratory exchange variables were continuously recorded to determine peak VO_2_, peak VCO_2_, peak HR, and peak power (W_Peak_). The FS [13] and the CR10 [14] were also applied throughout the test to familiarize the participants with the scales. The four subsequent visits included one vigorous-intensity continuous exercise (VICE) used as a control condition and three HIIE exercise protocols performed in a counterbalanced order. The exercise protocols were performed at the same average intensity (70% of peak power—W_Peak_) and total duration (20 min), with different amplitudes. In this sense, a VICE session was conducted at 70% W_Peak_ as previously described [3]. This strategy was adopted to equalize the exercise conditions into the heavy exercise domain [15], allowing us to establish the relationship between HIIE stimulus and recovery intensities and the affective valence. All sessions were conducted under similar environmental conditions in the laboratory (humidity ≈ 60% and temperature ≈ 22 °C). The physiological (respiratory variables and HR) and psychological (FS and RPE) variables were recorded during the exercise sessions. Figure 1 shows the experimental design of the study.

### 2.3. Procedures

#### 2.3.1. Anthropometric Measurements

The body mass and height were measured using a weighing scale with a stadiometer (Welmy 110 CH, Welmy, SP, Brazil) to determine body mass index (BMI). The chest, abdomen, and thigh skinfold thicknesses were measured (Slim Guide, Rosscraft Innovations, Inc., Vancouver, BC, Canada) and used to estimate body density (Jackson & Pollock, 1978) and body fat percentage (Siri, 1961). The technical procedures followed the ACSM [11] recommendations.

#### 2.3.2. Psychological Variables

The affective valence was measured using the FS [13]. The FS is a bipolar scale that ranges from −5 (very bad) to +5 (very good), with zero as “neutral”. Other verbal anchors include −3 (bad), −1 (fairly bad), +1 (fairly good), and +3 (good). The Portuguese version of the FS presented high reproducibility [16]. In addition, the Category Ratio Scale (CR10) was used to measure the RPE [14]. The participants received instructions regarding the scale proposal and responses.

#### 2.3.3. Physiological Variables

The VO_2_, VCO_2_, and HR were continuously recorded during the experimental sessions (every 10 s). The respiratory variables were measured using a gas analyzer (CORTEX, Biophisik GmbH, Leipzig, Germany) calibrated before each test according to the manufacturer’s instructions. The HR was measured using a HR monitor (RS800CX, Polar Electro OY, Kempele, Finland).

#### 2.3.4. Maximal Exercise Test

Before the beginning of the test, participants’ blood pressure and HR were measured after a 5 min resting period in a supine position. The maximal exercise test was performed on a cycle ergometer using an initial power output of 50 W. The power output increased by 30 W every 2 min until volitional fatigue [17]. During the test, participants were asked to maintain 70 to 90 rpm according to their preference. The variation in cadence was allowed considering that the ergometer modifies the resistance to maintain the power output depending on the rpm. The power output attained at the end of the last completed stage of the test was defined as the W_Peak_, and the linear interpolation method was used for participants who reached volitional fatigue before completing the stage. During the test, respiratory variables and HR were continuously recorded to determine VO_2Peak_, VCO_2Peak_, and maximal HR (HR_Max_). The FS and CR10 were applied at the end of every stage of the maximal exercise test. The last three values of VO_2_ and VCO_2_ were averaged to determine the peak value. The HR_Max_ was the highest value observed in the last 30 s of the test.

#### 2.3.5. Experimental Sessions

The exercise sessions were performed at the same average intensity (70% of W_Peak_) and a total duration of 20 min. The HIIE sessions presented the following configurations: (a) HIIE-90/50 = 10 × (1 min at 90% W_Peak_/1 min at 50% W_Peak_); (b) HIIE-100/40 = 10 × (1 min at 100% W_Peak_/1 min at 40% W_Peak_); (c) HIIE-110/30 = 10 × (1 min at 110% W_Peak_/1 min at 30% W_Peak_). The three HIIE sessions were performed using a stimulus–recovery ratio of 1:1, but with different amplitudes (HIIE-90/50 = 57%; HIIE-100/40 = 86%; HIIE-110/30 = 114%). The 70% W_Peak_ was chosen due to its compatibility with a vigorous intensity [11]. On the basis of the average intensity, variations of HIIE amplitude were proposed [3].

Before the exercise sessions, participants’ blood pressure and HR were measured, and the FS was applied 5 min prior. Then, a 3 min warmup was performed [18] at 30% W_Peak_ to provide the same metabolic disturbance for all participants. During the exercise sessions, respiratory variables and HR were recorded every 10 s. Regarding the psychological variables, the FS and CR10 were applied every minute immediately before the stimulus or recovery change. After the exercise sessions, the FS was recorded at 5, 10, and 15 min.

#### 2.3.6. Statistical Analysis

The participants’ characteristics are presented as the mean and standard deviation. The mean values of variables were calculated for each minute of exercise sessions, resulting in a total of 20 data points for each variable (VO_2_, VCO_2_, HR, CR10, and FS) in each condition (VICE, HIIE-90/50, HIIE-100/40, and HIIE-110/30). The physiological variables were converted to relative data (i.e., percentage of peak). Repeated-measures ANOVA was used to compare the mean values of each variable across exercise conditions. Mauchly’s sphericity test was conducted, and the Greenhouse–Geisser correction was applied when necessary. Then, linear correlations between the intensity variables (%VO_2Peak_, %VCO_2Peak_, %HR_Max_, and CR10) and the FS for each condition were calculated using the Pearson correlation coefficient. The significance level was established at 5% (*p* ≤ 0.05). The Pearson correlation coefficient was interpreted as follows: null (r < 0.5), low (0.5 < r < 0.7), moderate (0.7 < r < 0.9), and high (r ≥ 0.9) [19]. The data may be positive or negative depending on the nature of the correlation. The analyses were performed using the SPSS^®^ 25.0 for Windows (SPSS, Inc., Chicago, IL, USA).

## 3. Results

### 3.1. General Characteristics

The participants’ characteristics are presented in Table 1. Regarding the comparison of the recorded variables, significant main effects for the condition were found for the %VO_2Peak_ [F_(1.085, 20.610)_ = 7.407; *p* = 0.011; ηP^2^ = 0.280; ε = 0.362], %VCO_2Peak_ [F_(1.415, 20.610)_ = 23.745; *p* < 0.001; ηP^2^ = 0.556; ε = 0.472], %HR_Max_ [F_(3, 57)_ = 226.241; *p* < 0.001; ηP^2^ = 0.556; ε = 0.417], CR10 [F_(1.164, 22.120)_ = 6.219; *p* = 0.017; ηP^2^ = 0.247; ε = 0.388], and FS [F_(1.678, 31.888)_ = 19.869; *p* < 0.001; ηP^2^ = 0.511; ε = 0.559]. The mean and standard deviation values for each variable, as well as post hoc values to identify specific differences between conditions, are presented in Table 2.

### 3.2. Correlations

Null to high correlations were found across the exercise conditions and variables, as shown in Figure 2. All intensity variables presented significant inverse low and high correlations in VICE, HIIE-90/50, and HIIE-100/40. For the HIIE-110/30 condition, significant moderate and high inverse correlations were observed only for %HR_Max_ and CR10, respectively. However, nonsignificant null correlations were observed for %VO_2Peak_ and %VCO_2Peak_, indicating that these variables were unrelated to the affective valence in HIIE-110/30. The %HR_Max_ and CR10 presented higher inverse correlations with FS, indicating that both internal load variables could be used as markers for affective valence. In addition, %VO_2Peak_ and %VCO_2Peak_ presented low-to-moderate inverse correlations for VICE, HIIE-90/50, and 100/40 conditions.

## 4. Discussion

The objective of the present study was to verify the dose–response relationship between exercise intensity and affective valence in three HIIE sessions performed at the same average intensity and total time, but with different amplitudes. The main finding of the present study was the high inverse correlation between RPE and affective valence in different HIIE configurations. Previous studies [5,6,7] showed similar results; however, all studies performed only one HIIE session. In this sense, the present study could establish a dose–response relationship between HIIE intensities and affective valence using the RPE.

The present study measured exercise intensity using four internal load variables (%VO_2Peak_, %VCO_2Peak_, %HR_Max_, and CR10). Regarding HR and RPE, moderate-to-high correlations were found for all exercise conditions. The RPE presented similar results to previous studies [5,7]. RPE showed higher correlation values than the other internal load variables, which was previously explained by the cognitive processes involved in RPE and affect [5]. In this sense, this study adds that the inverse relationship between RPE and affect is observed in several HIIE sessions, and that the RPE is sensible to identify differences in amplitude even when the average intensity is the same across the sessions. Therefore, the dose–response relationship between HIIE intensity and affective valence may be established using RPE. The dual-mode theory is based on the premise that cognitive and interoceptive factors have different weights on individuals’ affective responses depending on the exercise intensity, below or above the lactate and ventilatory thresholds—LT/VT [20]. HIIE sessions generally comprise stimulus and recovery intensities above and below the LT/VT, respectively, with different durations that may also lead to different metabolic responses [8]. Therefore, considering the complex interactions between HIIE variables, it is impossible to establish a physiological marker reflecting the affective valence as previously established for continuous exercise [2].

Previous studies investigated the effect of the different stimuli and recovery durations on affective valence in work-matched HIIE sessions and showed that exposure to longer work periods resulted in lower affective valence [21,22]. The present study showed that the exposure to higher stimulus intensity in HIIE work-matched sessions also resulted in significantly lower affective valence (Table 2), while the CR10 presented significantly higher responses in HIIE-110/30, indicating higher perceived intensity in this exercise condition. The results mentioned above show the dose–response relationship between HIIE and affective valence, and the present study revealed that RPE might be used to identify this relationship. Similar results were also shown in a longitudinal study in which the relationship between affective valence and the RPE relative to the anaerobic threshold was higher than the relationship between affective valence and the speed relative to the anaerobic threshold [23].

No relationship between HR and FS was previously observed [5,7]. Ramalho Oliveira, Viana, Pires, Junior Oliveira, and Santos [5] showed low correlations between the %HR reserve and the FS for continuous (R^2^ = 0.07; r = 0.26) and HIIE (R^2^ = 0.05; r = 0.22) sessions. Farias-Junior, Browne, Astorino, and Costa [7] divided the HIIE session into three parts (beginning, middle, and end) and presented low correlations between %HR_Max_ and FS during the HIIE stimulus (beginning r = −0.11, middle r = −0.12, and end r = −0.27) and recovery periods (beginning r = −0.16, middle r = −0.32, and end-r = −0.35). These controversial results could be explained by the HIIE configuration [5] or by differences in physical activity level [7]. In addition, both studies grouped data according to the exercise duration in quintiles [5] and (approximately) in tertiles [7]. In contrast, the present study included the data point of each moment individually in the statistical analysis, which may also explain the differences in observed results. Despite these discrepancies, the present study indicated that HR is related to affective responses in HIIE. For example, a comparison between long and short HIIE sessions showed significantly higher HR and lower FS responses in the long HIIE session compared to the short one [24]. Although this was not a correlational study [24], an inverse response between the variables was observed.

Concerning the respiratory variables, the VO_2_ reflects the aerobic demand of the exercise [25], while the VCO_2_ reflects the sum of the following factors: aerobic metabolism, the bicarbonate buffering of H^+^ that comes from acid lactic, and hyperventilation of pulmonary capillary blood [12]. Therefore, while VO_2_ is a marker for aerobic metabolism, VCO_2_ may also be considered a marker for anaerobic metabolism. According to the dual-mode theory [26], previous studies showed that affective valence has an intrinsic inverse relationship with LT/VT [2,27]. This pattern is well established for continuous exercise [2,27], especially if we consider that the dual-mode theory [26] was proposed with a focus on the exercise of continuous nature. However, the intermittent nature of HIIE may affect this relationship. For example, Martinez, Kilpatrick, Salomon, Jung, and Little [22] performed three HIIE work-matched conditions with 24 min, varying the duration of bouts (30 s/30 s, 60 s/60 s, and 120 s/120 s) and found a more negative affect for the 120 s/120 s condition. This result suggests that the same stimulus–recovery ratio (1:1) across conditions was insufficient to maintain the same affective valence. Therefore, it is possible to hypothesize that the necessary amount of rest time for sufficient recovery (under the affective perspective) has an exponential pattern concerning the stimulus time. In this sense, Faelli et al. [28] compared a long [4 *×* (4 min at 90% W_Peak_/3 min at 30% W_Peak_)] versus a short [25 *×* (30 s at 100% W_Peak_/30 s at 20% W_Peak_)] HIIE in rowers and found higher VO_2_ and VCO_2_ responses in long HIIE, indicating that the time exposed to the work periods may present a significant impact on physiological responses. Corroborating to this premise, the comparison of three HIIE sessions with different stimulus (10 s, 30 s, and 60 s) and recovery (15 s, 45 s, and 90 s) durations showed that the longer stimulus duration resulted in higher metabolic responses and lower affective valence, despite the isoenergetic approach used to equalize the HIIE sessions [29]. The results presented in this study [29] show that the recovery duration could not be a linear function of the stimulus duration in HIIE. Unlike the abovementioned studies, we manipulated the stimulus and recovery intensity. Despite the other independent HIIE variables (i.e., duration bouts) controlled in previous studies [22,28,29], our results seem to make sense, considering that we found higher %VCO_2Peak_ responses in HIIE-110/30 compared to the other HIIE sessions, which could be explained by the exposure to the higher stimulus intensity (instead of duration) and, consequently, anaerobic demand. This result may indicate that not only the rest duration but also rest intensity could not be linearly reduced to provide sufficient recovery for affective valence increase.

In the present study, we found a low correlation between affective valence and respiratory variables in HIIE-110/30. Similar results were previously shown by Ramalho Oliveira, Viana, Pires, Junior Oliveira, and Santos [5], who proposed an exhausting HIIE configuration with a 2 min stimulus duration at an intensity of 100% of VO_2Peak_. Roloff et al. [30] demonstrated that the affective valence is intrinsically related to VO_2_ in four HIIE sessions configured using the critical power. The authors stated that the severity of homeostatic disturbance would guide the affective valence [30]. Therefore, the poor correlation between affect and respiratory variables in the HIIE-110/30 could be explained by the homeostatic disturbance that occurs in supramaximal intensities independently of respiratory responses, such as the increase in muscle H^+^ concentration or the decrease in muscle ATP, phosphocreatine, and pH [8]. Therefore, it is possible to assume that respiratory variables may be associated with the affective valence in HIIE sessions in which the anaerobic demand is not too high, such as the HIIE “long intervals” [31] in which the stimulus intensity is near maximal (i.e., 90–100% of VO_2Peak_ velocity). However, it should be noted that the use of too long a stimulus time as proposed for long intervals (≥60 s) of HIIE [31] may also negatively influence the affective responses [22].

### Limitations

Some limitations of the present study should be considered. The power calculation was conducted according to the analysis used in the first study of the major project [3]. Therefore, the small sample size should be considered in interpreting our results. Considering the relationship between psychological outcomes and physical activity habits [32], the lack of information regarding the physical activity history of participants could influence the interpretation of our results. In addition, this study did not include other models of interval exercise, such as sprint interval exercise. Therefore, these results could not be extrapolated to this model of interval exercise. In this sense, future studies should investigate whether the results presented here may be applied to the sprint interval exercise.

## 5. Conclusions

In conclusion, a dose–response relationship with HIIE intensity could be established only using the HR or, preferably, RPE for the HIIE prescription in young men. Therefore, recommendations for HIIE configuration should consider RPE as an alternative variable to manipulating HIIE prescription, especially when the affective valence is a target variable in the exercise program. From a practical perspective, considering the possible influence of affective responses on exercise adoption and maintenance, exercisers and coaches may use the RPE for HIIE prescription in order to precisely control the affective valence when performing HIIE sessions. Despite the findings of the present study, future studies should test this relationship for other populations.

## Figures and Tables

**Figure 1 ijerph-20-02698-f001:**
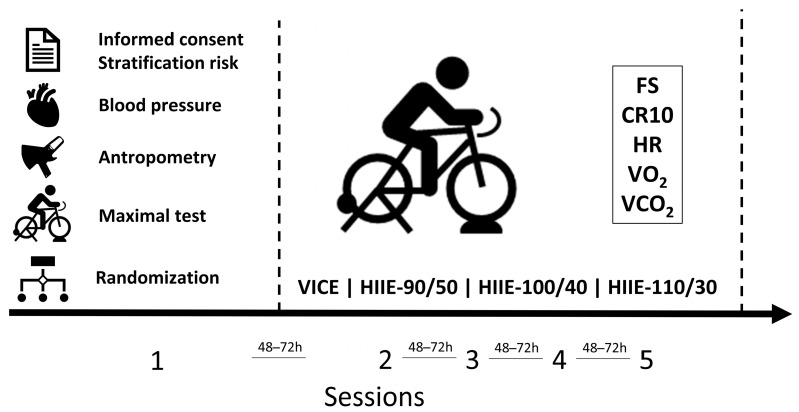
Experimental design. FS—Feeling Scale; CR10—Category Ratio Scale; HR—heart rate; VO_2_—oxygen consumption; VCO_2_—carbon dioxide output.

**Figure 2 ijerph-20-02698-f002:**
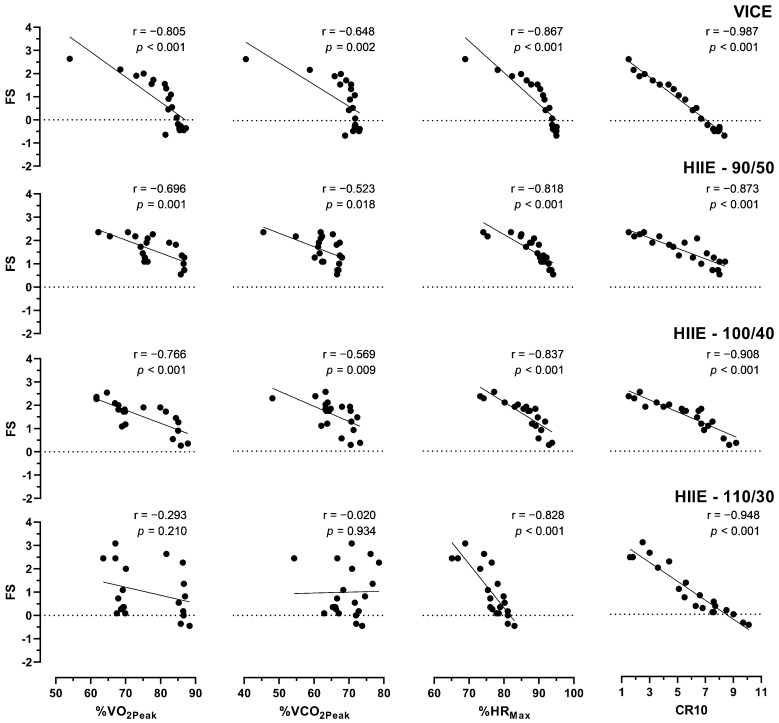
Correlations between intensity variables and Feeling Scale.

**Table 1 ijerph-20-02698-t001:** Participants’ characteristics.

Variables	M	SD	CI_95%_	
			Lower	Upper
Age (years)	24.6	3.9	21.9	27.3
Height (m)	1.74	0.06	1.70	1.78
Body mass (kg)	72.7	7.0	68.0	77.4
BMI (kg·m^−2^)	24.0	2.4	22.4	25.6
Percentage body fat	9.9	4.7	7.6	13.8
VO_2Peak_ (mL·kg^−1^·min^−1^)	46.0	6.9	41.3	50.6
VCO_2Peak_ (mL·kg^−1^·min^−1^)	53.0	7.4	48.6	57.4
HR_Max_ (bpm)	184.7	6.9	180.0	189.3
Peak power (W·kg^−1^)	3.23	0.46	2.91	3.53

M—mean; SD—standard deviation; CI—confidence interval; BMI—body mass index; HR—heart rate; RCP—respiratory compensation point.

**Table 2 ijerph-20-02698-t002:** Mean and standard deviation values of psychophysiological variables.

Variables	VICE		HIIE-90/50		HIIE-100/40		HIIE-110/30		Post-Hoc
	M	SD	M	SD	M	SD	M	SD	
%VO_2Peak_	80.4	7.8	78.1	7.3	74.3	8.8	76.2	9.2	HIIE-100/40 < all
%VCO_2Peak_	68.3	7.3	62.9	5.3	65.7	5.6	69.7	5.5	HIIE-90/50 < all; HIIE-100/40 < HIIE-110/30
%HR_Max_	89.6	6.7	88.0	5.5	86.0	5.8	76.4	4.8	HIIE-110/30 < HIIE-100/40 < HIIE-90/50, VICE
CR10	5.4	2.2	5.4	2.2	5.5	2.2	6.0	2.5	HIIE-110/30 > all
FS	0.76	1.05	1.56	0.58	1.55	0.65	0.99	1.09	VICE, HIIE-110/30 < HIIE-90/50, HIIE-100/40

%VO_2Peak_—percentage of peak oxygen consumption; %VCO_2Peak_—percentage of peak carbon dioxide output; %HR_Max_—percentage of maximal heart rate; CR10—category ratio scale; FS—Feeling Scale; M—mean; SD—standard deviation.

## Data Availability

Not applicable.

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
