# Peer review of "Is There a Dose–Response Relationship between High-Intensity Interval Exercise (HIIE) Intensity and Affective Valence? Analysis of Three HIIE Sessions Performed with Different Amplitudes"

_ijerph, 2023, doi:10.3390/ijerph20032698_

Round 1

Reviewer 1 Report

Dear authors:

Thanks for submitting this interesting and specific article. In general, the article is very easy to read and well written. No concerns about the use of English language have been detected.

Nevertheless, some comments could help to improve the article.

Abstract:

L-16: Some more details should be given in the abstract concerning the characteristics of the participants. Anyway, the sample of participants seems quite reduced. Please add information on this.

L-30: Some practical implications should be added at the end of the abstract.

L-33 – 39: The whole section “0.How to Use This Template” should be deleted.  

Introduction:

L-43: Is it ethically appropriate to self-cite several articles of one of the authors? Please consider this aspect when working on the resubmission.

Materials and methods

Participants

The study is well designed, and sufficient details are given on the procedures, but: …

L-86:  Eleven men sere included. How many persons were excluded?

L-87:  “should be” between …????

It would be important to add more information about the characteristics of the participants. Physical activity history? What kind of sports do they practice or practiced in the past? Influence of PA on results? Pleasure /dislike ?

L-97: Category-Rate Scale (CR10)

L-143: Kempele (Finland)

Discussion:

In the discussion section, several times results are repeated which already had been detailed in the Results section. It is not necessary to do this so extensively here.

Limitations:

I consider that it could be interesting to ‘open’ a specific limitations section for this article as there should be more details on this issue and some practical implications should be presented.

References:

Please have also a look at: https://www.frontiersin.org/articles/10.3389/fpsyg.2022.825738/full

Author Response

Reviewer 1

Dear authors:

Thanks for submitting this interesting and specific article. In general, the article is very easy to read and well written. No concerns about the use of English language have been detected.

Nevertheless, some comments could help to improve the article.

Authors: Thank you for your comments. We hope to fully meet your concerns.

Abstract:

L-16: Some more details should be given in the abstract concerning the characteristics of the participants. Anyway, the sample of participants seems quite reduced. Please add information on this.

Authors: We did not record more information regarding the characteristics of participants. The only information we had on this respect is the VO2Peak already reported in the table 2 of the manuscript. However, we included this issue as a limitation in the discussion section considering the limited space in the abstract.

L-30: Some practical implications should be added at the end of the abstract.

Authors: We included a sentence with a recommendation for HIIE prescription. Please, check in the manuscript file.

L-33 – 39: The whole section “0.How to Use This Template” should be deleted.

Authors: Done. Thank you for your comment.

Introduction:

L-43: Is it ethically appropriate to self-cite several articles of one of the authors? Please consider this aspect when working on the resubmission.

Authors: Thank you for your comment. Two references were excluded.

  1. Oliveira, B. R., et al. (2015). "Differences in exercise intensity seems to influence the affective responses in self-selected and imposed exercise: a meta-analysis." Front Psychol 6: 1105.
  2. Oliveira, B. R., et al. (2013). "Continuous and high-intensity interval training: which promotes higher pleasure?" PLoS One 8(11): e79965.

Materials and methods

Participants

The study is well designed, and sufficient details are given on the procedures, but: …

L-86:  Eleven men were included. How many persons were excluded?

Authors: One individual was excluded for fail in complete the experimental visits. We included this information in the manuscript.

L-87:  “should be” between …????

Authors: We changed the sentence. Now the sentence is “Participants aged 18 - 35 years old and at low risk for cardiovascular disease [11], physically active or not, were included in the study.”

It would be important to add more information about the characteristics of the participants. Physical activity history? What kind of sports do they practice or practiced in the past? Influence of PA on results? Pleasure /dislike?

Authors: We did not record this information. The only information we had in this respect is the VO2Peak already reported in table 2 of the manuscript. However, we included this issue as a limitation in the discussion section.

L-97: Category-Rate Scale (CR10)

Authors: Done.

L-143: Kempele (Finland)

Authors: Done.

Discussion:

In the discussion section, several times, results are repeated, which already had been detailed in the Results section. It is not necessary to do this so extensively here.

Authors: We excluded the following sentence.

“Regarding HR and RPE, moderate-to-high correlations were found for all exercise conditions.”

The sentence below was also changed.

“The present study showed that the exposure to higher stimulus intensity in HIIE work-matched sessions also resulted in significantly lower affective valence (Table 2), while the CR10 presented significantly higher responses in HIIE-110/30, indicating higher perceived intensity in this exercise condition”

The new sentence is:

“The present study showed similar findings, but with higher perceived intensity in HIIE-110/30. Thus, the present study revealed that RPE might be used to identify this relationship.

The other results presented in the discussion section were obtained from other studies to discuss with our results.

Limitations:

I consider that it could be interesting to ‘open’ a specific limitations section for this article as there should be more details on this issue and some practical implications should be presented.

Authors: We included a “Limitation” section.

References:

Please have also a look at: https://www.frontiersin.org/articles/10.3389/fpsyg.2022.825738/full

Authors: Thank you for your comment. We included a sentence regarding this study in the discussion section.

Similar results were also shown in a longitudinal study in which the relationship between affective valence and the RPE relative to the anaerobic threshold was higher than the relationship between affective valence and the speed relative to the anaerobic threshold [23].

Reviewer 2 Report

First at all I would like to congratute authors for their work on this paper. I found it very interesting. 

The paper follows a correct scientific structure.

The introduction provides a good background and the methodology is well described.

The results are correctly shown but table 2 could be clearer adding  post hoc results as superscripts, and including a paragraph explaning the differences shown. 

The discussion has a good extension. It clearly compares their results with other studies, including a final paragraph with the study limitations. 

The conclusion is clear and supported by the results but it shouldnt generalise, the sample only included eleven young men. 

I just made some minor comments in the PDF attached that could improve the quality of the paper. 

Author Response

Reviewer 2

First at all I would like to congratulate authors for their work on this paper. I found it very interesting.

The paper follows a correct scientific structure.

The introduction provides a good background and the methodology is well described.

Authors: Thank you for your comments.

The results are correctly shown but table 2 could be clearer adding  post hoc results as superscripts, and including a paragraph explaning the differences shown.

Authors: We made the changes in table 2. Please, check the table in the manuscript file.

The discussion has a good extension. It clearly compares their results with other studies, including a final paragraph with the study limitations. The conclusion is clear and supported by the results but it shouldnt generalise, the sample only included eleven young men.

Authors: Thank you for your comments. We included a sentence in the conclusion section regarding your concern.

I just made some minor comments in the PDF attached that could improve the quality of the paper.

Authors: We made the changes asked, and we hope to fully meet your expectations.
